# Dietary fiber intake and cognitive impairment in older patients with chronic kidney disease in the United States: A cross-sectional study

Feiyan Li[1,2☯], Hongxi Chen[3☯], Nan Mao[3]*, Hong Liu[1,2]*

1 School of Clinical Medicine, Chengdu Medical College, Chengdu, Sichuan, China, 2 Department of Experiment Teaching Center of Clinical Medicine, The First Affiliated Hospital of Chengdu Medical College, Chengdu, Sichuan, China, 3 Department of Nephrology, The First Affiliated Hospital of Chengdu Medical College, Chengdu, Sichuan, China

☯ These authors contributed equally to this work.
* hong@cmc.edu.cn (HL); maonanlyb@163.com (NM)

## Abstract

### Background

High-fiber diet has been associated with better cognitive performance. However, the association between dietary fiber intake and cognition in older patients with chronic kidney disease (CKD) remains unknown. Hence, this study aimed to investigate the effect of dietary fiber intake on cognition in older patients with CKD.

### Methods

This study included participants aged ≥60 years who provided data on social demography, cognitive tests (Consortium to Establish a Registry for Alzheimer's disease Word Learning [CERAD-WL], CERAD Delayed Recall [CERAD-DR], Animal Fluency Test [AFT], and Digit Symbol Substitution Test [DSST]), diet, and other potential cognition-related variables from the National Health and Nutrition Examination Survey 2011–2014. Fully-adjusted multivariate logistic regression subgroup models were performed, and multiple linear regression analyses were employed to examine the association between dietary fiber intake and cognition in patients with CKD.

### Results

A total of 2461 older adults were included, with 32% who suffered from CKD. Participants with CKD scored lower in CERAD-WL, CERAD-DR, AFT, and DSST. Patients with CKD consuming low dietary fiber (≤25 g/day) had a higher risk of CERAD-WL and DSST impairments. High dietary fiber intake eliminated the differences in CERAD-WL and DSST impairments between the CKD and non-CKD participants. However, no associations were observed between CKD and CERAD-DR and AFT impairments regardless of dietary fiber intake. A positive linear relationship between dietary fiber intake and AFT score was observed in older patients with CKD.

**Data Availability Statement:** Publicly available datasets are available online for this study. The repository/repositories name and accession

numbers are available online at http://www.cdc.gov/nchs/nhanes.htm

**Funding:** This research was funded by Sichuan Applied Psychology Research Center (No. CSXL-202B11) and Health Commission of Sichuan Province (No. 20PJ172). The funder had no role in study design, data collection and analysis, decision to publish, or preparation of the manuscript.

**Competing interests:** The authors have declared that no competing interests exist.

## Conclusion

High dietary fiber intake may benefit cognitive function in older patients with CKD. High-fiber diet management strategies could potentially mitigate cognitive impairment in this group of patients.

## Introduction

Approximately 10% of adults worldwide suffer from some form of chronic kidney disease (CKD), causing 1.2 million deaths each year, and it is projected to become the fifth leading cause of death globally by 2040 [1]. As a systemic disease, CKD also involves the central nervous system and is complicated by cognitive impairment [2]. CKD is considered one of the strongest risk factors for cognitive impairment, which refers to comprehensive deficits of brain processes for learning, memory, and sensory function that varies in severity, from moderate cognition decline to severe dementia [3]. A substantial proportion of patients with CKD suffer from cognitive impairment, even in the early stages of the disease [4], and the prevalence of CKD and cognitive impairment increase with age [5].

Albuminuria and estimated glomerular filtration rate (eGFR) are the extensively researched risk factors for cognitive impairment in CKD. While studies have consistently shown that albuminuria is associated with worse cognitive performance [6, 7], the effect of eGFR on cognitive impairment remains unclear. A large prospective study in older adults showed that a declined eGFR was associated with the incidence of dementia independent of stroke even in participants with baseline eGFR $\geq$60 mL/min/1.73 m$^2$ [8]. In addition, numerous factors, including age, education level, CKD duration, hypertension, diabetes, hemoglobin, albumin, glycemia, serum parathyroid hormone, uric acid, acidosis, electrolyte disorders, poor nutrition/protein energy wasting, disturbed sleep, and polypharmacy were reported as CKD-specific risk factors for cognitive impairment [9–12].

Despite medical advancements, there has been very limited success in the improvement of cognitive impairment in CKD. However, in recent years, the significance of dietary or nutritional changes in older adults with cognitive impairment has received great attention[13, 14]. A study suggested that low fiber intake is associated with cognitive impairments [15]. Studies suggested that high dietary fiber intake can help improve some aspects of cognitive function in the general adult population and the hypertensive population aged >60 years [16, 17].

A high-fiber, plant-dominant and low-protein diet, which reportedly modulates the gut microbiome, reduces uremic toxin, controls uremia without renal replacement therapy, and enhances cardiovascular health, has been proposed in CKD [1]. However, at present, studies investigating the effect of dietary fiber intake on the cognitive function of patients with CKD are scarce. Given that older adults are at a higher risk for cognitive impairment compared to their younger counterparts, the association between dietary fiber intake and cognition in older adults with and without CKD in the United States (US) was evaluated using data obtained from National Health and Nutrition Examination Survey (NHANES) 2011–2014. Based on the stratification of fiber intake and CKD status among the participants, we hypothesized that high dietary fiber intake contributes to reduced cognitive impairment in patients with CKD. We believe that these findings can provide insights into reduced cognition levels in CKD.

## Materials and methods

### Study participants

The data of participants in this cross-sectional study were obtained from NHANES 2011–2014, which is a stratified, multistage survey conducted in US civilian, non-institutionalized population. The study protocol was approved by the National Center for Health Statistics Institutional Review Board (https://www.cdc.gov/nchs/nhanes/irba98.htm) and all participants provided written informed consents upon the application to the NHANES. The secondary data analysis did not require additional institutional review board approval. This study recruited adults aged ≥60 years with complete cognitive function test data in the 2011–2014 survey cycles. The participants with incomplete data on CKD status, dietary fiber intake, and other covariates were excluded.

### CKD status

In early-stage CKD, the urinary albumin:creatinine ratio (UACR) is ≥30 mg/g; however, the eGFR remains normal. With CKD progression and the accumulation of uremic toxins, eGFR falls below 60 ml/min/ 1.73m$^2$. eGFR and UACR thresholds are usually used to define CKD [5]. In the present study, we defined CKD based on the above standards by the test value of UACR and the calculated value of eGFR. eGFR was estimated using the Chronic Kidney Disease Epidemiology Collaboration (CKD-EPI) 2021 creatinine-based equation recommended by the National Kidney Foundation and the American Society of Nephrology [18].

### Cognitive function

A series of cognitive function assessments, including word learning and recall modules from the Consortium to Establish a Registry for Alzheimer's Disease (CERAD), Animal Fluency test (AFT), and Digit Symbol Substitution test (DSST), were performed in 2011–2014 NHANES among participants aged ≥60 years. The CERAD tests were used to assess immediate and delayed learning ability for new verbal information [19]; there were three immediate recalls (CERAD-WL) and a delayed recall (CERAD-DR). In the immediate recalls, participants were instructed to read 10 unrelated words. Immediately following the presentation of the words, participants were asked to recall as many words as possible three times. After completing the AFT and DSST tests, delayed word recall occurred (approximately 8–10 min from the start of the word learning trials). The maximum score possible on each recall was 10. The AFT investigates categorical verbal fluency, a component of executive function [20]. Participants were asked to name as many animals as possible in 1 min and a point was given for each named animal. The total scores ranged from 1 to 40. The DSST was used to evaluate the abilities of processing speed, sustained attention, and working memory. In the DSST, nine numbers paired with symbols were provided, and the participants were required to copy the corresponding symbols in the 133 boxes that adjoin the numbers in 2 min. The score was the total number of correct matches.

In the above cognitive function tests, higher scores indicated better cognitive function. In accordance with the criteria for cognitive impairment in the previous literature, the lowest quartile of the CERAD-WL, CERAD-DR, AFT, and DSST scores were used as the cutoff points. As cognitive function declines with age, the cutoff points for different age groups were calculated (Table 1) [17,21]. The participants with scores below or equal to the cutoff points of the corresponding age group were defined as having cognitive impairment.

**Table 1. Cutoff points of cognitive function test scores adjusted by age.**

| Cognitive test | Cut off points | | |
|---|---|---|---|
| | ≥60y | ≥70y | ≥80y |
| Immediate recall (CERAD-WL) score | 17 | 16 | 14 |
| Delayed recall (CERAD-DR) score | 5 | 4 | 3 |
| Verbal fluency (AFT) score | 14 | 12 | 12 |
| Executive function & processing speed (DSST) score | 38 | 33 | 28 |

**Abbreviations**: CERAD-WL, Consortium to Establish a Registry for Alzheimer's Disease Word Learning; CERAD-DR, Consortium to Establish a Registry for Alzheimer's Disease Delayed Recall; AFT, Animal Fluency test; DSST, Digit Symbol Substitution test.

## Measurement of dietary data

The dietary data of participants were obtained from the NHANES database through two 24-h recall surveys. The first 24-h dietary recall was conducted in the mobile examination center, and the second dietary recall proceeded over the phone 3–10 days later. The dietary nutrients were calculated as the average of data from both times if those data were available. Otherwise, the first 24-h dietary recall was used. As the main dietary components, protein and carbohydrate intakes were reported to associated with cognition in previous studies [13,14]. In this study, in addition to dietary fiber intake, we included energy, protein, and carbohydrate as covariates.

The European Food Safety Authority and the UK Scientific Advisory Committee on Nutrition recommend a daily fiber intake of 25 g/day and 30 g/day for adults, respectively [22]. The American Heart Association recommends a dietary fiber intake of at least 25 g/day for adults [23]. In this study, we divided the participants into two groups based on the fiber intake value of 25 g/day: low-level group (≤25 g/day) and high-level group (>25 g/day).

## Covariates

Potential cognition-related covariates, including age, sex, ethnicity, education level, marital status, smoking status, body mass index, hypertension, diabetes, coronary heart disease, stroke, malignancy, depression, sleep disorder, UACR, eGFR, hemoglobin, albumin, blood urea nitrogen, creatinine, uric acid, total dietary intake of energy, carbohydrates, and protein, were analyzed in this study.

Race/ethnicity was classified as Mexican American, non-Hispanic white, non-Hispanic black, or other races. Educational level was categorized as <9 years, 9–12 years, and >12 years. Marital status was classified as married/living with a partner or widowed/divorced/separated/never married. Smoking status was defined as never smokers (smoked<100 cigarettes), current smokers, and former smokers (quit smoking after smoking>100 cigarettes). Disease history (such as hypertension, diabetes, coronary heart disease, stroke, malignancy, and sleep disorder) was determined by self-report of whether the disease was diagnosed by a physician in the past. Depression status was based on mental health questionnaire (DPQ020). Total energy, carbohydrates, and protein intakes were obtained from the participants' 24-h nutritional information. In addition, the kidney conditions questionnaire (KIQ025) was used to determine whether the participants received dialysis treatment.

## Statistical analyses

The Kolmogorov–Smirnov normality test was performed to determine the normality of the distribution of continuous variables. Mean ± standard deviation was used for normal variables, and median (interquartile range) for non-normally distributed variables. Categorical variables were expressed as absolute numbers with percentages. Differences in continuous values with a normal distribution were compared using an independent samples t-test. For non-normally distributed data, the groups were compared using the Mann–Whitney U test. The chi-square test was performed to compare categorical variables. We investigated the difference in cognitive scores and cognitive impairments between CKDs and non-CKD participants. A multivariate logistic regression subgroup analysis was performed to analyze the odds ratios (OR) and 95% confidence intervals (95% CIs) of cognitive impairment between low and high dietary fiber intake groups and in participants with and without CKD. The models were fully adjusted by age, sex, ethnicity, education level, marital status, smoking status, body mass index, hypertension, diabetes, coronary heart disease, stroke, malignancy, depression, sleep disorder, UACR, eGFR, hemoglobin, albumin, blood urea nitrogen, creatinine, uric acid, total dietary intake of energy, carbohydrates, and protein. The interaction between CKD status and dietary fiber intake was inspected by the likelihood ratio test. Stratified analyses of the relationship between dietary fiber intake and cognitive impairment were performed, including the following variables: sex, marital status (married/living with a partner vs. living alone), education level ($\leq$12 years vs. >12 years), smoking status (never smoking vs. Current/former smoker), and history of hypertension and diabetes (absence vs. presence). Multiple linear regression analyses were performed to examine the association between dietary fiber intake and cognition in patients with CKD. Furthermore, a restricted cubic spline (RCS) with four knots was performed to assess the potential nonlinear relationship between dietary fiber intake and cognitive impairment. To assess the robustness of our findings, we performed sensitivity analyses using fully-adjusted logistic regression subgroup analysis after excluding participants with extreme fiber intake <4.7g/day (2.5%) or >38.7 g/day (97.5%). In addition, considering the possible effect of dialysis treatment on cognitive impairment, patients with CKD on dialysis treatment were excluded from the sensitivity analyses. All statistical analyses were performed using the statistical software packages R 4.1.1 (http://www.R-project.org) and Free Statistics software version 1.7 [24]. A two-tailed $p$-value <0.05 was considered statistically significant.

# Results

## Study population

Overall, 2934 adults aged $\geq$60 years with complete cognitive function tests from the 2011–2014 survey cycles of NHANES were included. Participants with incomplete CKD status (n = 204), dietary fiber intake (n = 192), and other covariates data (n = 77) were excluded. A total of 2461 participants were included in the final analysis (Fig 1).

## Baseline characteristics

The baseline characteristics of the participants according to CKD status are shown in Table 2. Of 2461, 787 (32%) participants had CKD. The average age was 69.3 (6.7) years and 49.7% were male. Individuals with CKD were older; had a lower education level; were widowed, divorced, separated, or never married; were current or former smokers; had a higher incidence of hypertension, diabetes, coronary heart disease, stroke, and depression; and had lower consumption of energy, protein, carbohydrate, and fiber.

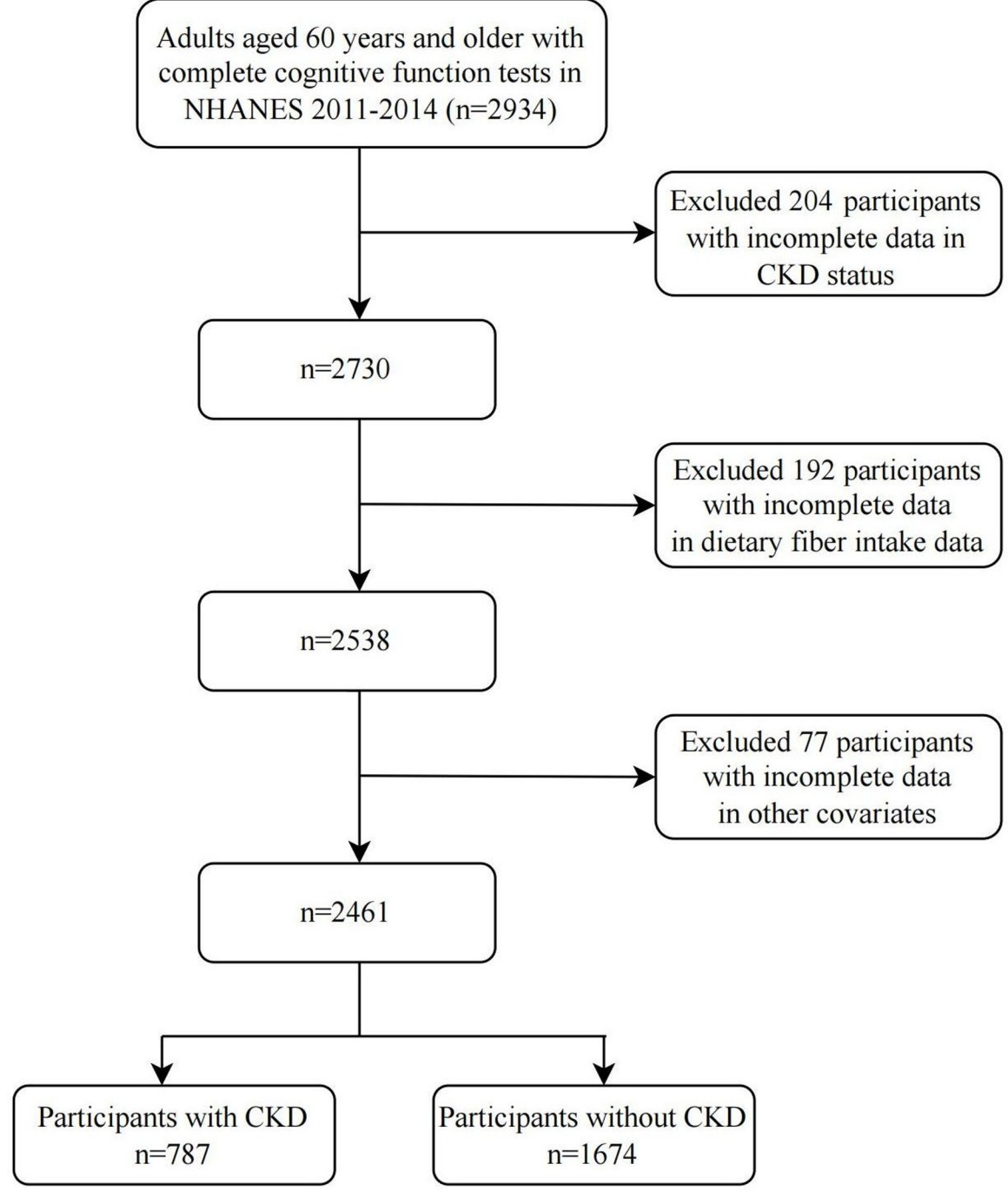

**Fig 1. Flow diagram of the study. Abbreviation**: NHANES, National Health and Nutrition Examination Survey; CKD, chronic kidney disease.

### Cognitive function of the participants with and without CKD

Table 3 shows the cognitive function of the participants. Participants with CKD showed lower scores in all four cognitive function tests (CERAD-WL, CERAD-DR, AFT, and DSST) as compared to those without CKD ($p < 0.001$). Cognitive impairment was determined when the

**Table 2. Characteristics of NHANES 2011-2014 US participants aged ≥60 years.**

| Characteristics | Total (n = 2461) | Non-CKD (n = 1674) | CKD (n = 787) | P-value |
|---|---|---|---|---|
| Sex, n (%) | | | | 0.848 |
| Male | 1222 (49.7) | 829 (49.5) | 393 (49.9) | |
| Female | 1239 (50.3) | 845 (50.5) | 394 (50.1) | |
| Age, years | 69.3 ± 6.7 | 68.1 ± 6.3 | 71.8 ± 6.9 | < 0.001 |
| Ethnicity, n (%) | | | | < 0.001 |
| Mexican | 215 (8.7) | 165 (9.9) | 50 (6.4) | |
| Non-Hispanic White | 1231 (50.0) | 821 (49) | 410 (52.1) | |
| Non-Hispanic Black | 557 (22.6) | 345 (20.6) | 212 (26.9) | |
| Others | 458 (18.6) | 343 (20.5) | 115 (14.6) | |
| Education level, years, n (%) | | | | < 0.001 |
| <9 | 259 (10.5) | 166 (9.9) | 93 (11.8) | |
| 9–12 | 331 (13.4) | 198 (11.8) | 133 (16.9) | |
| >12 | 1871 (76.0) | 1310 (78.3) | 561 (71.3) | |
| Marital status, n (%) | | | | < 0.001 |
| Married/Living with partner | 1455 (59.1) | 1031 (61.6) | 424 (53.9) | |
| Widowed/Divorced/Separated/Never married | 1006 (40.9) | 643 (38.4) | 363 (46.1) | |
| Smoking, n (%) | | | | 0.01 |
| Never | 1209 (49.1) | 855 (51.1) | 354 (45) | |
| Current | 312 (12.7) | 196 (11.7) | 116 (14.7) | |
| Former | 940 (38.2) | 623 (37.2) | 317 (40.3) | |
| Body Mass Index, kg/m$^2$ | 29.1 ± 6.3 | 28.9 ± 6.2 | 29.4 ± 6.6 | 0.056 |
| Hypertension, n (%) | | | | < 0.001 |
| Absence | 934 (38.0) | 737 (44) | 197 (25) | |
| Presence | 1527 (62.0) | 937 (56) | 590 (75) | |
| Diabetes, n (%) | | | | < 0.001 |
| Absence | 1913 (77.7) | 1389 (83) | 524 (66.6) | |
| Presence | 548 (22.3) | 285 (17) | 263 (33.4) | |
| Coronary Heart Disease, n (%) | | | | < 0.001 |
| Absence | 2234 (90.8) | 1562 (93.3) | 672 (85.4) | |
| Presence | 227 (9.2) | 112 (6.7) | 115 (14.6) | |
| Stroke, n (%) | | | | < 0.001 |
| Absence | 2303 (93.6) | 1592 (95.1) | 711 (90.3) | |
| Presence | 158 (6.4) | 82 (4.9) | 76 (9.7) | |
| Malignancy, n (%) | | | | 0.084 |
| Absence | 1979 (80.4) | 1362 (81.4) | 617 (78.4) | |
| Presence | 482 (19.6) | 312 (18.6) | 170 (21.6) | |
| Depression, n (%) | | | | 0.014 |
| Absence | 1886 (76.6) | 1307 (78.1) | 579 (73.6) | |
| Presence | 575 (23.4) | 367 (21.9) | 208 (26.4) | |
| Sleep disorder, n (%) | | | | 0.132 |
| Absence | 2162 (87.9) | 1482 (88.5) | 680 (86.4) | |
| Presence | 299 (12.1) | 192 (11.5) | 107 (13.6) | |
| Urinary Albumin: Creatinine Ratio, mg/g | 70.5 ± 429.8 | 9.6 ± 6.0 | 200.0 ± 743.9 | < 0.001 |
| Estimated glomerular filtration rate, ml/min/1.73/m$^2$ | 76.0 ± 18.9 | 83.4 ± 12.3 | 60.3 ± 20.7 | < 0.001 |
| Hemoglobin, g/dL | 13.8 ± 1.4 | 14.0 ± 1.3 | 13.4 ± 1.5 | < 0.001 |
| Albumin, g/L | 42.0 ± 3.0 | 42.2 ± 2.9 | 41.6 ± 3.1 | < 0.001 |
| Blood Urea Nitrogen, mmol/L | 5.7 ± 2.4 | 5.1 ± 1.6 | 7.1 ± 3.2 | < 0.001 |

*(Continued)*

**Table 2.** (Continued)

| Characteristics | Total (n = 2461) | Non-CKD (n = 1674) | CKD (n = 787) | P-value |
|---|---|---|---|---|
| Creatinine, umol/L | 88.4 ± 38.7 | 77.3 ± 15.8 | 112.1 ± 57.7 | < 0.001 |
| Uric Acid, umol/L | 337.6 ± 85.3 | 321.5 ± 76.5 | 371.7 ± 92.7 | < 0.001 |
| Dietary energy, kcla | 1822.5 ± 679.6 | 1880.2 ± 682.7 | 1700.0 ± 656.6 | < 0.001 |
| Dietary protein, g | 72.9 ± 29.5 | 75.4 ± 30.0 | 67.7 ± 27.9 | < 0.001 |
| Dietary carbohydrate, g | 222.2 ± 86.8 | 229.1 ± 87.7 | 207.4 ± 82.8 | < 0.001 |
| Dietary fiber, g | 17.0 ± 9.0 | 18.0 ± 9.5 | 15.0 ± 7.5 | < 0.001 |

**Abbreviation**: NHANES, National Health and Nutrition Examination Survey.

scores were lower than the cutoff points adjusted by age. The incidence of cognitive impairment was higher in the CKD group.

## Dietary fiber intake affects the association between CKD and cognitive impairment

As shown in Fig 2, there was an inverse association between dietary fiber intake and cognitive impairment in the CKD group after adjusting for potential confounders. Compared to non-CKD participants, CKD participants consuming low dietary fiber ($\leq$25g/day) tended to have a higher risk of CERAD-WL and DSST impairments. The adjusted OR values for the CKD group in CERAD-WL and DSST impairments were 1.38 (95% CI: 1.06–1.80, $p$ = 0.018) and 1.41 (95% CI: 1.04–1.92, $p$ = 0.028), respectively. In the high dietary fiber intake group, the differences in CERAD-WL and DSST impairments between the CKD and non-CKD groups were not significant. In addition, there were no associations between CKD and CERAD-DR

**Table 3. Cognitive function of US older adults in NHANSE 2011–2014.**

| Cognitive tests | Total (n = 2461) | Non-CKD (n = 1674) | CKD (n = 787) | P-value |
|---|---|---|---|---|
| CERAD-WL score | 19.1 ± 4.5 | 19.6 ± 4.4 | 18.2 ± 4.7 | < 0.001 |
| CERAD-DR score | 6.0 ± 2.3 | 6.2 ± 2.2 | 5.6 ± 2.4 | < 0.001 |
| AFT score | 16.9 ± 5.5 | 17.4 ± 5.5 | 15.7 ± 5.2 | < 0.001 |
| DSST score | 47.0 ± 17.0 | 49.4 ± 16.9 | 41.8 ± 16.1 | < 0.001 |
| CERAD-WL impairment, n (%) | | | | < 0.001 |
| Absence | 1765 (71.7) | 1240 (74.1) | 525 (66.7) | |
| Presence | 696 (28.3) | 434 (25.9) | 262 (33.3) | |
| CERAD-DR impairment, n (%) | | | | 0.004 |
| Absence | 1758 (71.4) | 1226 (73.2) | 532 (67.6) | |
| Presence | 703 (28.6) | 448 (26.8) | 255 (32.4) | |
| AFT impairment, n (%) | | | | < 0.001 |
| Absence | 1763 (71.6) | 1242 (74.2) | 521 (66.2) | |
| Presence | 698 (28.4) | 432 (25.8) | 266 (33.8) | |
| DSST impairment, n (%) | | | | < 0.001 |
| Absence | 1817 (73.8) | 1299 (77.6) | 518 (65.8) | |
| Presence | 644 (26.2) | 375 (22.4) | 269 (34.2) | |

**Abbreviations**: NHANES, National Health and Nutrition Examination Survey; CERAD-WL, Consortium to Establish a Registry for Alzheimer's Disease Word Learning; CERAD-DR, Consortium to Establish a Registry for Alzheimer's Disease Delayed Recall; AFT, Animal Fluency test; DSST, Digit Symbol Substitution test.

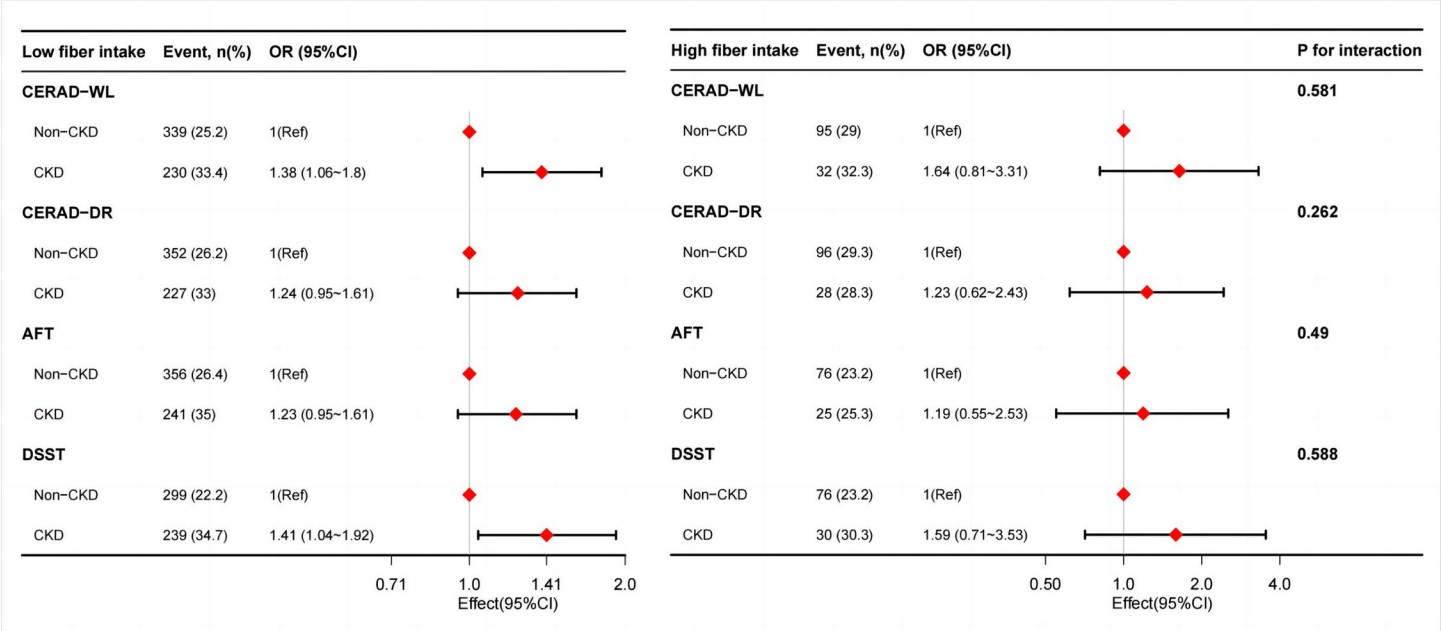

**Fig 2. Association between CKD and cognitive function impairment in older adults with different dietary fiber intake levels in the US.** The models were adjusted by age, sex, ethnicity, education, marital status, smoking, body mass index, hypertension, diabetes, coronary heart disease, stroke, malignancy, depression, sleep disorder, UACR, eGFR, hemoglobin, albumin, blood urea nitrogen, creatinine, uric acid, dietary energy, dietary protein, and dietary carbohydrate. **Abbreviations**: CKD, chronic kidney disease; UACR, urinary albumin:creatinine ratio; eGFR, estimated glomerular filtration rate; CERAD-WL, Consortium to Establish a Registry for Alzheimer's Disease Word Learning; CERAD-DR, Consortium to Establish a Registry for Alzheimer's Disease Delayed Recall; AFT, Animal Fluency test; DSST, Digit Symbol Substitution test.

and AFT impairments regardless of dietary fiber intake. There were no significant interactions by fiber intake on the association between CKD and cognition in any subgroup.

## Stratified analyses on the association between dietary fiber intake and cognitive impairment

Stratified analyses were performed to assess the potential effect of dietary fiber intake on cognitive impairment in the subgroups. Given the effects of different dietary fiber intakes on CERAD-WL and DSST impairments, the two types of cognitive impairment were analyzed in the subgroups stratified by sex, marital status, education level, smoking status, and history of hypertension and diabetes. In the low dietary fiber intake group, CERAD-WL (Fig 3) or DSST impairment (Fig 4) were prominent in CKD patients who were male, living alone, had a higher education level, never smoked, had diabetes, and had no hypertension. No difference in CERAD-WL or DSST impairment was observed between CKD and non-CKD patients in the high fiber intake group.

## Association between dietary fiber intake and cognition in older patients with CKD

The multiple linear regression analyses showed that dietary fiber intake was positively correlated to the AFT score (β = 0.147, $p$ = 0.001) after adjustment for the full set of covariates. A linear relationship between dietary fiber intake and AFT score was observed in older patients with CKD (Fig 5). No associations were found between dietary fiber intake and CERAD-WL (β = 0.046, $p$ = 0.324), CERAD-DR (β = 0.073, $p$ = 0.119), and DSST (β = 0.049, $p$ = 0.229)

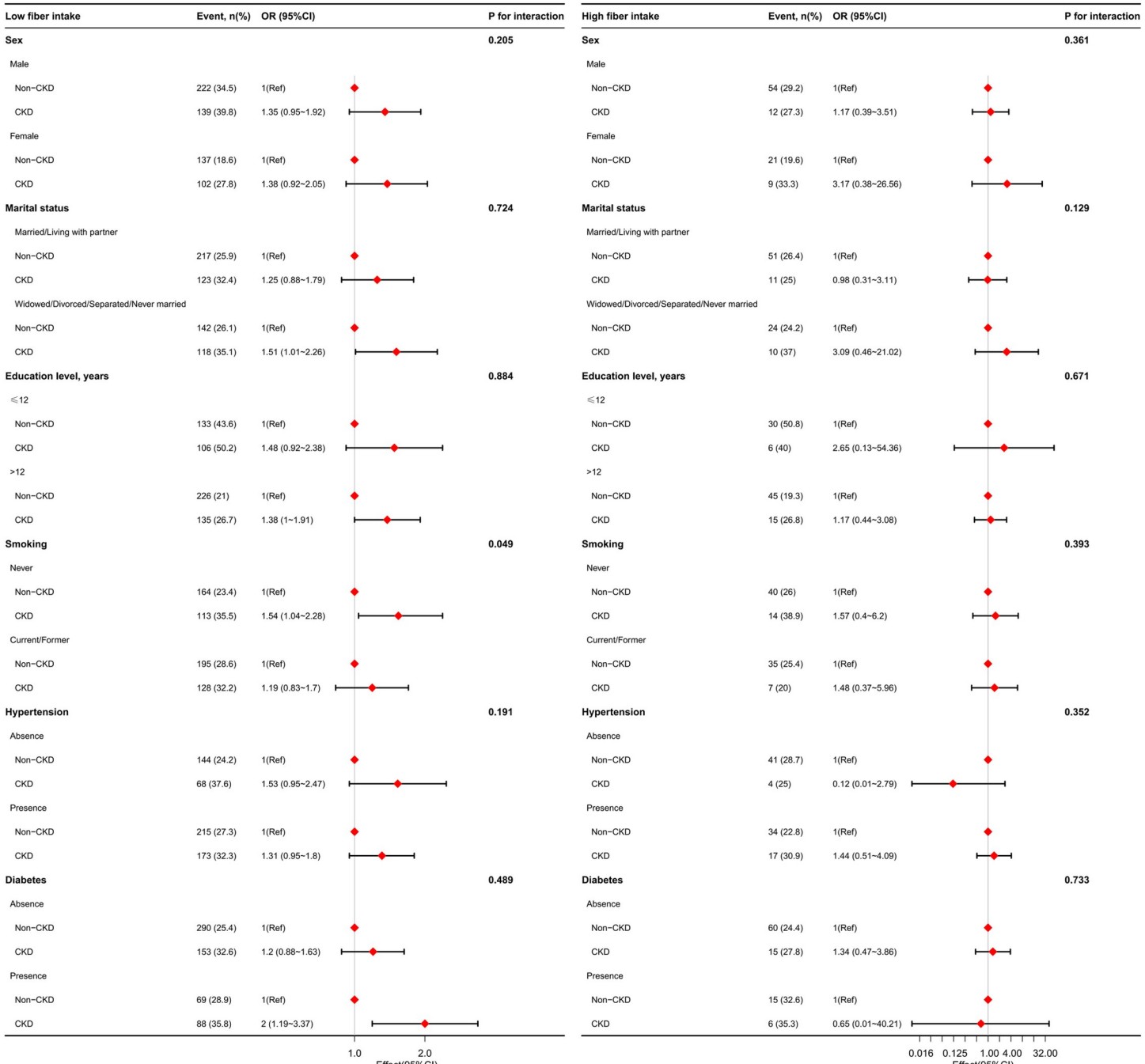

**Fig 3. Stratified analysis of the relationship between CKD and CERAD-WL impairment in older adults with different dietary fiber intake levels in the US.** The models were adjusted by age, sex, ethnicity, education, marital status, smoking, body mass index, hypertension, diabetes, coronary heart disease, stroke, malignancy, depression, sleep disorder, UACR, eGFR, hemoglobin, albumin, blood urea nitrogen, creatinine, uric acid, dietary energy, dietary protein, and dietary carbohydrate, except for the stratification factor itself. **Abbreviations:** CKD, chronic kidney disease; CERAD-WL, Consortium to Establish a Registry for Alzheimer's Disease Word Learning; UACR, urinary albumin:creatinine ratio; eGFR, estimated glomerular filtration rate.

scores. In addition, nonlinear relationships were not observed between dietary fiber intake and cognitive impairment (S1 Fig).

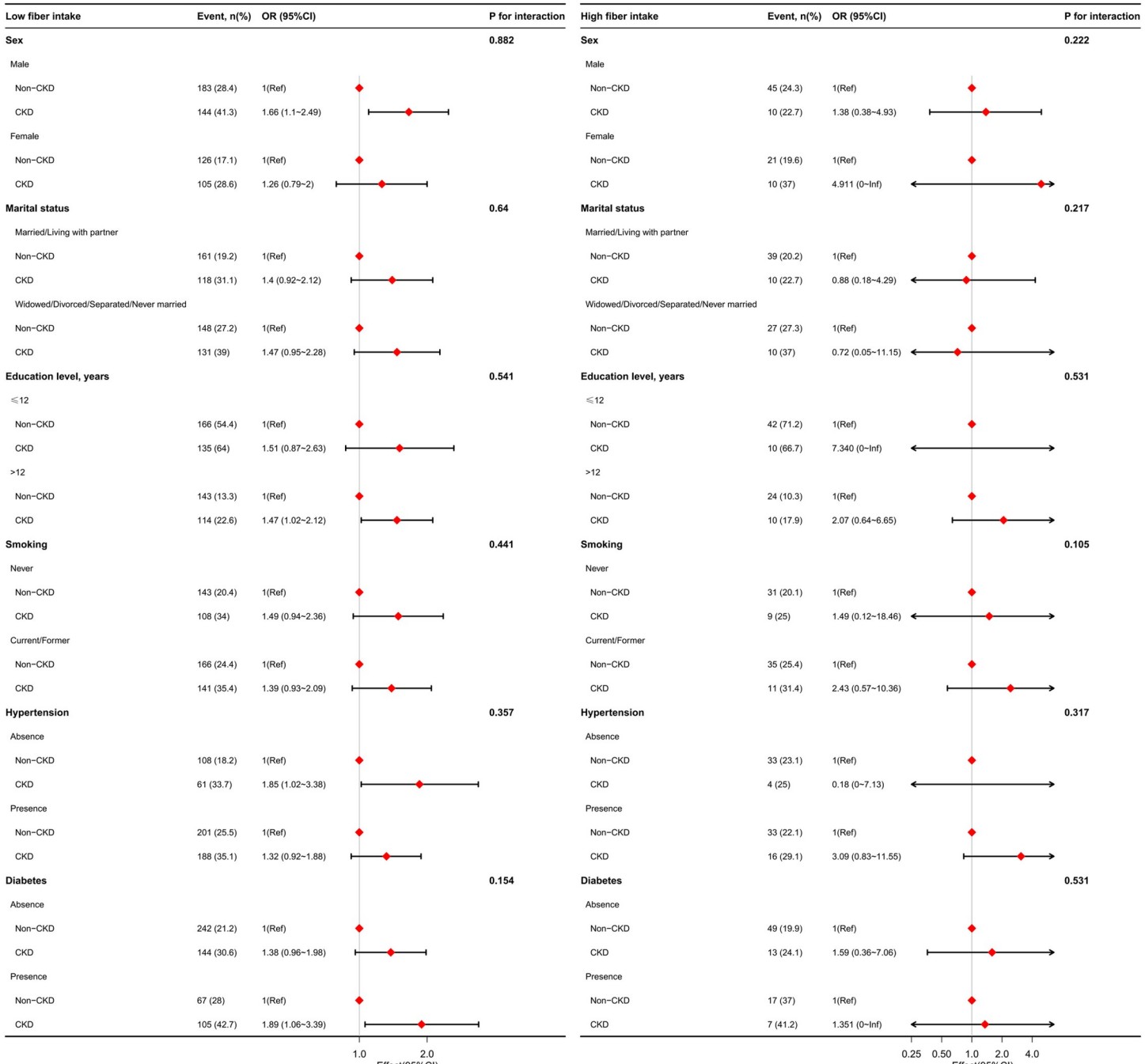

**Fig 4. Stratified analysis of the relationship between CKD with DSST impairment in older adults with different dietary fiber intake levels in the US.** The models were adjusted by age, sex, ethnicity, education, marital status, smoking, body mass index, hypertension, diabetes, coronary heart disease, stroke, malignancy, depression, sleep disorder, UACR, eGFR, hemoglobin, albumin, blood urea nitrogen, creatinine, uric acid, dietary energy, dietary protein, and dietary carbohydrate, except for the stratification factor itself. **Abbreviations**: CKD, chronic kidney disease; DSST, Digit Symbol Substitution Test; UACR, urinary albumin:creatinine ratio; eGFR, estimated glomerular filtration rate.

## Sensitivity analysis

Participants with dietary fiber intake <4.7g/day and >38.7g/day were excluded for sensitivity analysis. We found that the effect of dietary fiber intake on the relationship between cognitive

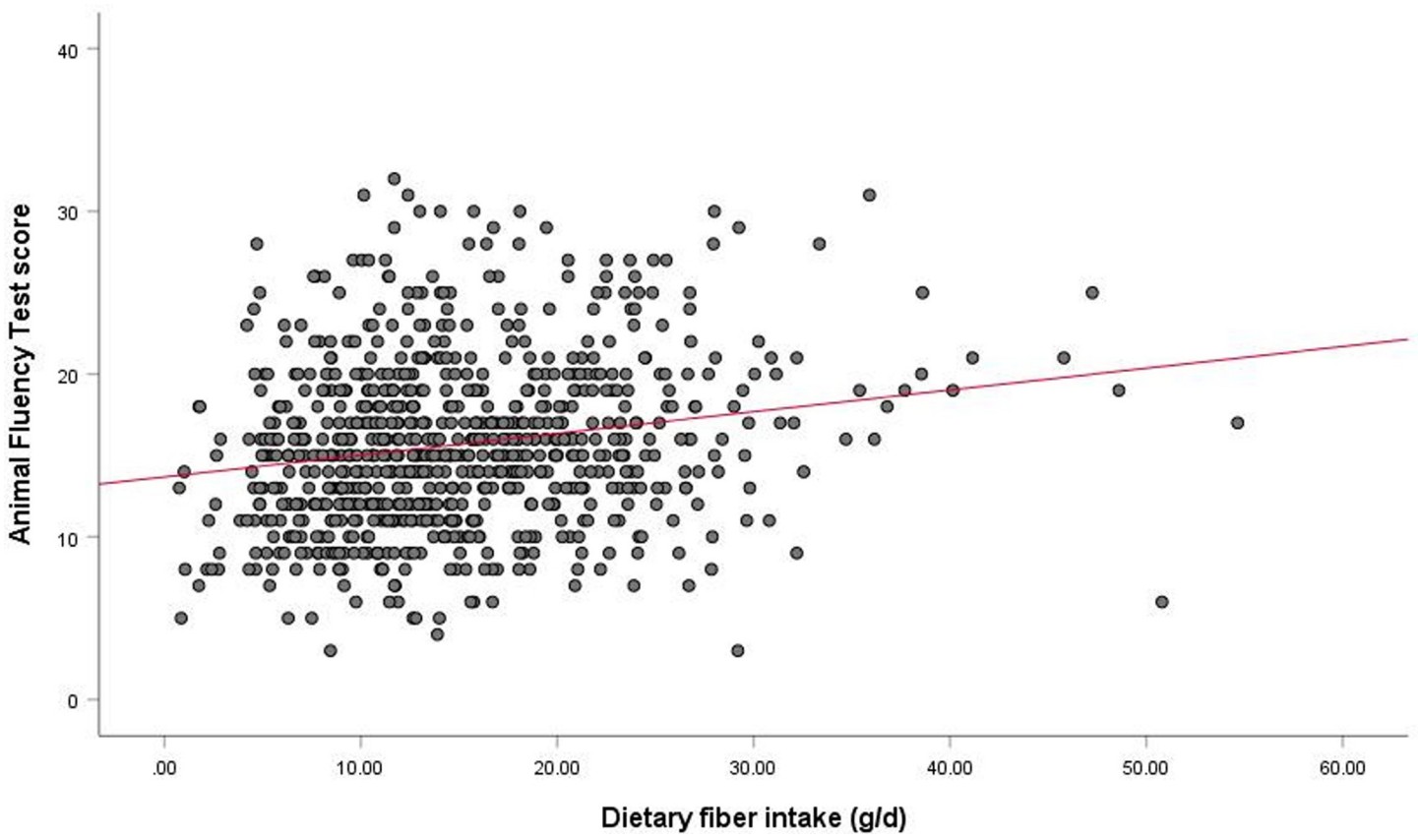

**Fig 5. Dose-response curve between dietary fiber intake and AFT score after adjusting for full covariates in older patients with CKD. Abbreviations**: AFT, Animal Fluency Test; CKD, chronic kidney disease.

impairment and CKD was stable (S1 Table). The results were also similar when CKD patients receiving dialysis treatment were excluded from the analysis (S2 Table).

## Discussion

This cross-sectional study assessed the cognitive function of older adults in the US by CERAD-WL, CERAD-DR, AFT, and DSST, which embodies immediate and delayed learning ability, executive function, processing speed, sustained attention, and working memory. The results indicated that global cognitive performance was worse in patients with CKD. CKD patients with low dietary fiber intake showed a higher risk of cognitive impairment in CERAD-WL and DSST as compared to those without CKD, implying the worse immediate learning ability, processing speed, sustained attention, and working memory of CKD patients. In the high-fiber diet group, the differences in cognitive impairment between the CKD and non-CKD participants were not significant. A positively linear relationship between dietary fiber intake and AFT score was observed in older patients with CKD. Despite the lack of robust data, high dietary fiber intake might benefit cognitive function in older patients with CKD.

The prevalence of cognitive impairment in patients with CKD ranged from 10% to an astonishing 87% [25–27], depending on the testing methods of cognitive assessment and CKD stage. In our study, the incidence of cognitive impairment in older patients with CKD was at 32%-35%. A similar result was found in a cross-sectional study that included 825 adults with CKD aged ≥55 years [27], which showed a 4%–28% prevalence of cognitive impairment; however, a

negative correlation was observed between eGFR and the prevalence of cognitive impairment. The slightly higher rate in our study may be attributed to the age differences between the study populations. Executive function and attention are most impaired in mild CKD, whereas in severe CKD, cognitive impairment is more global and more severe, which may be because individuals are older by the time they reach a severe stage of CKD [28]. A previous study based on NHANES III showed that moderate CKD patients aged 20–59 years were significantly related to poorer cognitive performance in visual attention and learning/concentration [29]. Specific to older patients with CKD, the participants scored lower in all four tests: CERAD-WL, CERAD-DR, AFT, and DSST. This suggested that CKD could influence immediate and delayed learning ability, executive function, processing speed, sustained attention, and working memory. A meta-analysis showed that patients with CKD performed worse than the control groups on Global Cognition tests, including Orientation & Attention, Memory, Language, Concept Formation & Reasoning, and Executive Function assessment, in line with our study [30]. Some potential mechanisms contributing to cognitive impairment in CKD were cerebrovascular diseases, particularly small-vessel cerebrovascular disease, and other CKD-specific risk factors, such as uremic metabolites, dialysis factors, anemia, and aluminum [31].

Furthermore, we found that CKD participants with low dietary fiber intake had a higher risk of CERAD-WL and DSST impairments, resulting in worse immediate learning ability, processing speed, sustained attention, and working memory than the non-CKD participants. However, high dietary fiber intake eliminates the differences between CKD and non-CKD participants, thus preventing older patients with CKD from cognitive impairment. A positive linear relationship was also found between dietary fiber intake and AFT score in patients with CKD, suggesting the benefit of high-fiber diets on cognition in CKD. Similarly, a result was also observed in the general older adults, whose consumption of dietary fiber was positively associated with DSST score and a plateau in DSST score was suggested at a dietary fiber intake of 34 g per day [16]. A plant-based diet characterized by high fiber and polyunsaturated fat intake and low saturated fat and protein intake is related to better cognitive performance on memory and executive function in US older adults [32]. Researchers also established a consistent effect of high dietary fiber intake on the association between hypertension and cognitive impairment [17]. In the past few decades, many studies have suggested that the Mediterranean diet (MedDiet) can reduce the risk of cardiovascular disease and improve cognition [33, 34]. The MedDiet includes plenty of vegetables, fruits, and cereals, which provide 33 g of dietary fiber per day [35]. On the contrary, a large cross-sectional study involving 48749 participants in the United Kingdom suggested that better cognitive ability was associated with moderate intake of fiber, not with high and low fiber intake. There was no reasonable explanation for the U-shaped relationship between cognition and fiber intake [36]. Considering the possibility of a non-linear relationship, we also need a large clinical study to validate our findings regarding the association between dietary fiber intake and cognition in CKD.

Emerging evidence has shown that diet plays a crucial role in shaping the gut microbiome and modulating the structure and function of the brain through the microbiota-gut-brain axis, which is made up of neuroendocrine, neural, and immune communication channels and has emerged as a key conduit for the effects of nutrition on the brain [37]. Reportedly, dietary fiber regulates the gut microbiome and provides the crucial substrate to the community of microbes in the distal gut. Dietary fiber is fermented by the gut microbiome in the intestines, and short-chain fatty acid (SCFA) is generated as a major metabolite, which is often considered a key candidate mediator of the microbiota-gut-brain axis and involved in psychological functioning [38]. Animal experiments proved that high-fiber intake in the maternal diet restores maternal obesity-induced cognitive and social deficits in offspring by reshaping the gut microbiome via the microbiota-gut-brain axis and SCFA in both mother and offspring mice [39]. On the

contrary, mice with long-term dietary fiber deficiency exhibited impaired cognition, gut microbiota dysbiosis, and reduced SCFA production [15]. A randomized, double-blind, placebo-controlled, crossover-design study demonstrated that supplementation with dietary fiber, polydextrose, resulted in a modest improvement in cognition by affecting gut-to-brain communication [40]. In short, SCFA and the microbiota-gut-brain axis are the major contributors and pathways to the benefit of a high-fiber diet on cognitive impairment.

This study has some important limitations that should be acknowledged. First, although NHANES included large representative health and nutrition data, this study showed the correlation between high fiber intake and improved cognition in older patients with CKD due to its cross-sectional nature. Nonetheless, verification of the causal relationship still requires a large longitudinal study. Second, the diagnostic criteria of CKD are structural or functional abnormalities of the kidney for at least 3 months. However, the values of serum creatinine and UACR were measured only once in each participant in NHANES 2011–2014. The duration of the disease could not be determined, which may lead to a slight overestimation of the prevalence of CKD. Reportedly, an estimated 47% of individuals aged > 70 years in the US are affected by CKD [41], which is similar to that of our study, suggesting that 32% of individuals aged ≥60 years had CKD. Third, according to a study, CKD-EPI 2021 eGFR equation with creatinine and cystatin C is more accurate than the equation with creatinine alone [42]. However, cystatin C was not examined in NHANES 2011–2014, which prevented us from estimating GFR more precisely. Therefore, we chose CKD-EPI 2021 creatinine-based equation to calculate GFR, which is also recommended by the National Kidney Foundation. Fourth, although numerous related covariates were adjusted in the regression models, perhaps some potential confounders were not excluded in this study.

In summary, we found that older patients with CKD are associated with cognitive impairment across multiple domains, especially in the immediate learning ability, processing speed, attention, and working memory after adjustment for numerous sociodemographic and clinical confounders. High dietary fiber intake might benefit the association between CKD and cognitive impairment in this group of patients. Nonetheless, future longitudinal studies are warranted to validate our findings and facilitate clinical interventions that implement high-fiber diet management strategies to mitigate cognitive impairment in older patients with CKD.

## Supporting information

**S1 Fig. Nonlinear relationship between dietary fiber intake and cognitive impairment in older patients with CKD.** Solid and dashed lines represent the predicted value and 95% confidence intervals. The models were fully adjusted by age, sex, ethnicity, education, marital status, smoking, body mass index, hypertension, diabetes, coronary heart disease, stroke, malignancy, depression, sleep disorder, UACR, eGFR, hemoglobin, albumin, blood urea nitrogen, creatinine, uric acid, dietary energy, dietary protein, and dietary carbohydrate. **Abbreviations**: CKD, chronic kidney disease; UACR, urinary albumin:creatinine ratio; eGFR, estimated glomerular filtration rate; CERAD-WL, Consortium to Establish a Registry for Alzheimer's Disease Word Learning; CERAD-DR, Consortium to Establish a Registry for Alzheimer's Disease Delayed Recall; AFT, Animal Fluency test; DSST, Digit Symbol Substitution test.
(TIF)

**S1 Table. CKD and cognitive function impairment by levels of fiber intake among older adults without extreme dietary fiber intake in the US, NHANES 2011–2014.** The logistic regression models were adjusted by age, sex, ethnicity, education, marital status, smoking, body mass index, hypertension, diabetes, coronary heart disease, stroke, malignancy, depression, sleep disorder, UACR, eGFR, hemoglobin, albumin, blood urea nitrogen, creatinine, uric

acid, dietary energy, dietary protein, and dietary carbohydrate. **Abbreviations:** CKD, chronic kidney disease; NHANES, National Health and Nutrition Examination Survey; UACR, urinary albumin:creatinine ratio; eGFR, estimated glomerular filtration rate; CERAD-WL, Consortium to Establish a Registry for Alzheimer's Disease Word Learning; CERAD-DR, Consortium to Establish a Registry for Alzheimer's Disease Delayed Recall; AFT, Animal Fluency test; DSST, Digit Symbol Substitution test.
(DOCX)

**S2 Table. CKD and cognitive function impairment by levels of fiber intake among older adults after excluding CKD patients receiving dialysis treatment in the US, NHANES 2011–2014.** The logistic regression models were adjusted by age, sex, ethnicity, education, marital status, smoking, body mass index, hypertension, diabetes, coronary heart disease, stroke, malignancy, depression, sleep disorder, UACR, eGFR, hemoglobin, albumin, blood urea nitrogen, creatinine, uric acid, dietary energy, dietary protein, and dietary carbohydrate. **Abbreviations:** CKD, chronic kidney disease; NHANES, National Health and Nutrition Examination Survey; UACR, urinary albumin:creatinine ratio; eGFR, estimated glomerular filtration rate; CERAD-WL, Consortium to Establish a Registry for Alzheimer's Disease Word Learning; CERAD-DR, Consortium to Establish a Registry for Alzheimer's Disease Delayed Recall; AFT, Animal Fluency test; DSST, Digit Symbol Substitution test.
(DOCX)

## Author Contributions

**Conceptualization:** Feiyan Li, Nan Mao, Hong Liu.

**Data curation:** Feiyan Li, Hongxi Chen.

**Funding acquisition:** Feiyan Li.

**Methodology:** Feiyan Li, Hongxi Chen.

**Project administration:** Nan Mao.

**Software:** Hongxi Chen.

**Supervision:** Hong Liu.

**Validation:** Nan Mao, Hong Liu.

**Writing – original draft:** Feiyan Li, Hongxi Chen.

**Writing – review & editing:** Nan Mao, Hong Liu.

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
