## [Decision Letter · Decision Letter 0]

2 Jul 2023

PONE-D-23-15009Dietary fiber intake and cognitive impairment in older patients with chronic kidney disease in US: a cross-sectional studyPLOS ONE

Dear Dr. liu,

Thank you for submitting your manuscript to PLOS ONE. After careful consideration, we feel that it has merit but does not fully meet PLOS ONE’s publication criteria as it currently stands. Therefore, we invite you to submit a revised version of the manuscript that addresses the points raised during the review process.

Please kindly address the comments of the reviewers if appropropriate

We look forward to receiving your revised manuscript.

Kind regards,

Yee Gary Ang, MBBS MPH

Academic Editor

PLOS ONE

Reviewers' comments:

Reviewer's Responses to Questions

**Comments to the Author**

1. Is the manuscript technically sound, and do the data support the conclusions?

Reviewer #1: Yes

Reviewer #2: Yes

2. Has the statistical analysis been performed appropriately and rigorously? 

Reviewer #1: I Don't Know

Reviewer #2: No

3. Have the authors made all data underlying the findings in their manuscript fully available?

Reviewer #1: Yes

Reviewer #2: Yes

4. Is the manuscript presented in an intelligible fashion and written in standard English?

Reviewer #1: No

Reviewer #2: No

5. Review Comments to the Author

Reviewer #1: The association of dietary fiber intake with cognition in older patients with chronic kidney disease (CKD) is unknown. The authors aimed to aim to investigate the effect of dietary fiber intake on cognition in older patients with CKD. They found that global cognitive performance was worse in the subjects with CKD than non-CKDs. They also found an association between dietary fiber intake and cognitive performance in older CKD patients.

1. Introduction; Reference 17 does not seem to support the sentence High dietary fiber intake is beneficial to improve some aspects of cognitive function in general adults and hypertensive populations over 60 years old.

2. At the end of Introduction, the sentence “Based on the stratification of fiber intake and CKD status among the participants, we hypothesized that high dietary fiber intake affects the association of CKD and cognitive function and contributes to reduced cognitive impairment in patients with CKD, which may provide some insight into cognition in CKD.” seems insufficient. A mechanism should be proposed, either here or in the Discussion part, for example, increased fibre intake shifts the gut microbiota towards reduced production of uraemic toxins.

3. The statement “We conducted a cross-sectional study.” appears problematic. The website of NHANES is better provided. The sentence “…written informed consents were obtained from participants before data collection.” appears problematic too, because the data were not collected by the authors.

4. The “Measurement of dietary data”, the authors should specify whether the data were collected by themselves or obtained from the NHANES database.

5. Discussion: As stated, executive function and attention are most impaired in mild CKD, while in severe CKD, cognitive impairment is more global and more severe, perhaps because individuals are older by the time they reach a severe stage of CKD [28]. Are the data from the present study consistent to those published in literature.

6. Reference 8 does not seem to support the sentence “A large prospective study in older adults showed a declined eGFR was associated with the incidence of dementia independent of stroke even in participants with baseline eGFR ≥60 mL/min/1.73 m2”.

7. Language：the manuscript is in need of English language editing, I read it myself and found many errors in English grammar, syntax, and word usage including, but are not limited to, lack of subject-verb agreement (singular vs. plural), inconsistent tense usage, improper word forms (noun vs. verb vs. adjective vs. gerund). Accordingly, we must insist that you have the text professionally edited by an English improvement service before the manuscript can be reconsidered for publication in the journal.

Reviewer #2: Authors investigated that whether the fiber intake of CKD patients effects the cognition. The topic is meaningful. However, the design of study is too simple to explain this question clearly. I recommend authors to re-design the analysis process. Besides, some major problems are listed here.

1. In the NHANES database, age of patients older than 80 years old were not provided. Did authors exclude this population?

2. How did author make the diagnosis of CKD? Because the NHANES database is a cross-sectional database. How did author get the information of kidney dysfunction over 3 months?

3. Subgroup analysis would help authors get more information.

4. Propensity score is recommended.

5. Maybe a non-linear association were existed between fiber intake and cognitive function. A dot-plot should be performed. Logistic analysis may not be suitable.

6. Subjects with dialysis history should be excluded. Because dialysis may also has an effect on the cognitive impairment.

6. PLOS authors have the option to publish the peer review history of their article (what does this mean?). If published, this will include your full peer review and any attached files.

Reviewer #1: No

Reviewer #2: No

---

## [Author Response · Author response to Decision Letter 0]

13 Jul 2023

Reviewer #1: The association of dietary fiber intake with cognition in older patients with chronic kidney disease (CKD) is unknown. The authors aimed to aim to investigate the effect of dietary fiber intake on cognition in older patients with CKD. They found that global cognitive performance was worse in the subjects with CKD than non-CKDs. They also found an association between dietary fiber intake and cognitive performance in older CKD patients.

1. Introduction; Reference 17 does not seem to support the sentence High dietary fiber intake is beneficial to improve some aspects of cognitive function in general adults and hypertensive populations over 60 years old.

Response：Thank you for pointing it out. We apologize that the references were not in order. Accordingly, we have arranged the references in order in the revised manuscript.

Reference 16

Prokopidis K, Giannos P, Ispoglou T, Witard OC, Isanejad M. Dietary Fiber Intake is Associated with Cognitive Function in Older Adults: Data from the National Health and Nutrition Examination Survey. Am J Med. 2022;135(8):e257-e62. https://doi.org/10.1016/j.amjmed.2022.03.022

Reference 17

Zhang H, Tian W, Qi G, Sun Y. Hypertension, dietary fiber intake, and cognitive function in older adults [from the National Health and Nutrition Examination Survey Data (2011-2014)]. Front Nutr. 2022;9:1024627. https://doi.org/10.3389/fnut.2022.1024627

2. At the end of Introduction, the sentence “Based on the stratification of fiber intake and CKD status among the participants, we hypothesized that high dietary fiber intake affects the association of CKD and cognitive function and contributes to reduced cognitive impairment in patients with CKD, which may provide some insight into cognition in CKD.” seems insufficient. A mechanism should be proposed, either here or in the Discussion part, for example, increased fibre intake shifts the gut microbiota towards reduced production of uraemic toxins. 

Response：Thank you for your valuable comment. Accordingly, we have revised the following sentences in the Introduction and Discussion sections of the manuscript.

Introduction (page 5, lines 66–68)

“A high-fiber, plant-dominant, and low-protein diet, which reportedly modulates the gut microbiome, reduces uremic toxin, controls uremia without renal replacement therapy, and enhances cardiovascular health, has been proposed in CKD [1].”

Discussion (page 18, lines 326–329)

“Dietary fiber is fermented by the gut microbiome in the intestines and short-chain fatty acid (SCFA) is generated as major metabolite, which is often considered a key candidate mediator of the microbiota-gut-brain axis and involved in psychological functioning [38].”

3. The statement “We conducted a cross-sectional study.” appears problematic. The website of NHANES is better provided. The sentence “…written informed consents were obtained from participants before data collection.” appears problematic too, because the data were not collected by the authors.

Response：Thank you for pointing it out. We have revised the following sentences in the Materials and Methods section accordingly.

Page 5, lines 79–80

“The data of participants in this cross-sectional study were obtained from NHANES 2011–2014, which is a stratified, multistage survey conducted in US civilian, non-institutionalized population.”

Page 5, lines 82–83

“… and all participants provided written informed consent upon the application to the NHANES.”

4. The “Measurement of dietary data”, the authors should specify whether the data were collected by themselves or obtained from the NHANES database.

Response：Thank you for your valuable suggestion. We have revised the sentence accordingly.

Page 7, lines 122–123

“The dietary data of participants were obtained from the NHANES database through two 24-h recall surveys.”

5. Discussion: As stated, executive function and attention are most impaired in mild CKD, while in severe CKD, cognitive impairment is more global and more severe, perhaps because individuals are older by the time they reach a severe stage of CKD [28]. Are the data from the present study consistent to those published in literature.

Response：Thank you for your question. We divided the CKD patients based on the eGFR value: low eGFR group (<30 ml/min/1.73 m2) and high eGFR group (≥30 ml/min/1.73 m2). There was no difference in age between both groups. AFT and DSST impairments were significant in the low eGFR group, suggesting that executive function, processing speed, sustained attention, and working memory were more impaired in severe CKD. This finding is consistent with previous studies. However, we were unable to establish the association between age and CKD severity, which may be attributed to the limited sample size. This point was not sufficiently addressed in our study because the main aim was to determine the association between dietary fiber intake and cognition.

6. Reference 8 does not seem to support the sentence “A large prospective study in older adults showed a declined eGFR was associated with the incidence of dementia independent of stroke even in participants with baseline eGFR ≥60 mL/min/1.73 m2”.

Response：Thank you for pointing it out. The references were not in order, and we apologize for the error. Accordingly, we have revised the following reference.

Reference 8

Singh-Manoux A, Oumarou-Ibrahim A, Machado-Fragua MD, Dumurgier J, Brunner EJ, Kivimaki M, et al. Association between kidney function and incidence of dementia: 10-year follow-up of the Whitehall II cohort study. Age Ageing. 2022;51(1). https://doi.org/10.1093/ageing/afab259

7. Language：the manuscript is in need of English language editing, I read it myself and found many errors in English grammar, syntax, and word usage including, but are not limited to, lack of subject-verb agreement (singular vs. plural), inconsistent tense usage, improper word forms (noun vs. verb vs. adjective vs. gerund). Accordingly, we must insist that you have the text professionally edited by an English improvement service before the manuscript can be reconsidered for publication in the journal.

Response：We appreciate your valuable suggestion. We regret that the quality of the language is not as per the journal’s standards. Accordingly, the manuscript has been edited for proper English language, grammar, punctuation, spelling, and overall style by a language editing service.

Reviewer #2: Authors investigated that whether the fiber intake of CKD patients effects the cognition. The topic is meaningful. However, the design of study is too simple to explain this question clearly. I recommend authors to re-design the analysis process. Besides, some major problems are listed here.

1. In the NHANES database, age of patients older than 80 years old were not provided. Did authors exclude this population?

Response：Thank you for your insightful question. In the NHANES database, individuals 80 and over are topcoded at 80 years of age. Therefore, we did not exclude this population, which was 383 in total.

2. How did author make the diagnosis of CKD? Because the NHANES database is a cross-sectional database. How did author get the information of kidney dysfunction over 3 months?

Response：Thank you for your questions. CKD was defined based on the UACR and calculated eGFR value, which was estimated using the Chronic Kidney Disease Epidemiology Collaboration 2021 creatinine-based equation recommended by the National Kidney Foundation and the American Society of Nephrology. However, the duration of the kidney disease could not be determined, which might lead to a slight overestimation of the prevalence of CKD. This was mentioned as a limitation of the study. Reportedly, an estimated 47% of individuals aged >70 years were affected by CKD in the US. A similar trend was observed in our study, and 32% of individuals were defined as CKD in the population aged ≥60 years.

3. Subgroup analysis would help authors get more information.

Response：Thank you for your valuable suggestion. Stratified analyses were performed, and the results and figures have been added to the article. In the low-dietary fiber intake group, CERAD-WL and DSST impairments were prominent in the CKD patients who were male, living alone, had a higher education level, never smoked, have diabetes, and have no hypertension. No difference was observed in the high-dietary fiber group.

4. Propensity score is recommended.

Response：Thank you for your suggestion. We attempted to perform a propensity score matching; however, the process resulted in the loss of a large number of samples. Hence, the analysis was not conducted. The fully adjusted models were used in the linear and logistic regression analysis, which could weaken the impact of confounding factors.

5. Maybe a non-linear association were existed between fiber intake and cognitive function. A dot-plot should be performed. Logistic analysis may not be suitable.

Response：Thank you for your suggestion. Dot-plot and dose-response curve were performed, and a linear relationship was observed between dietary fiber intake and AFT score in patients with CKD. The results have been included in the article. In addition, we attempted to analyze the non-linear association between dietary fiber intake and cognitive impairment using restricted cubic spline regression. In all participants, we found an L-shaped relationship between dietary fiber intake and cognitive impairment (Figure was attached below). However, in patients with CKD, we did not observe a non-linear association between dietary fiber and cognitive impairment, which may be due to the limited sample size. Considering the study focus on patients with CKD, we did not show this result.

6. Subjects with dialysis history should be excluded. Because dialysis may also has an effect on the cognitive impairment.

Response：Thank you for your valuable suggestion. The data on dialysis therapy showed that there were six CKD patients receiving dialysis treatment. The association between dietary fiber intake and cognitive impairment was analyzed after excluding CKD patients receiving dialysis treatment.

---

## [Decision Letter · Decision Letter 1]

7 Aug 2023

PONE-D-23-15009R1Dietary fiber intake and cognitive impairment in older patients with chronic kidney disease in the United States: A cross-sectional studyPLOS ONE

Dear Dr. liu,

Thank you for submitting your manuscript to PLOS ONE. After careful consideration, we feel that it has merit but does not fully meet PLOS ONE’s publication criteria as it currently stands. Therefore, we invite you to submit a revised version of the manuscript that addresses the points raised during the review process.

Please address the comments raised by the 2 interviewers. 

We look forward to receiving your revised manuscript.

Kind regards,

Yee Gary Ang, MBBS MPH

Academic Editor

PLOS ONE

Journal Requirements:

Reviewers' comments:

Reviewer's Responses to Questions

**Comments to the Author**

1. If the authors have adequately addressed your comments raised in a previous round of review and you feel that this manuscript is now acceptable for publication, you may indicate that here to bypass the “Comments to the Author” section, enter your conflict of interest statement in the “Confidential to Editor” section, and submit your "Accept" recommendation.

Reviewer #1: All comments have been addressed

Reviewer #3: (No Response)

2. Is the manuscript technically sound, and do the data support the conclusions?

Reviewer #1: Yes

Reviewer #3: Yes

3. Has the statistical analysis been performed appropriately and rigorously? 

Reviewer #1: Yes

Reviewer #3: Yes

4. Have the authors made all data underlying the findings in their manuscript fully available?

Reviewer #1: Yes

Reviewer #3: Yes

5. Is the manuscript presented in an intelligible fashion and written in standard English?

Reviewer #1: Yes

Reviewer #3: Yes

6. Review Comments to the Author

Reviewer #1: Abstract: the sentence "A total of 2461 older adults were included, of with 32% suffered from CKD" should be rephrased.

The language needs to be futher polished.

The sentence "Previous studies suggested that protein and carbohydrate intakes, which are the main dietary components, were associated with cognition [13,14]." should be rewritten for clarity.

Line 79, the word "peportedly" is inappropriately used.

Reviewer #3: Overall, I have no major criticisms on this ms; however, restricted cubic splines should be mentioned in the methods and the results should be provided in the text.

7. PLOS authors have the option to publish the peer review history of their article (what does this mean?). If published, this will include your full peer review and any attached files.

Reviewer #1: **Yes: **Hongliang Zhang

Reviewer #3: No

---

## [Author Response · Author response to Decision Letter 1]

16 Aug 2023

Reviewer #1: 

Abstract: the sentence "A total of 2461 older adults were included, of with 32% suffered from CKD" should be rephrased.

Response：Thank you for pointing it out. We have revised the sentence.

Page 2, lines 29

“A total of 2461 older adults were included, with 32% who suffered from CKD.”

The language needs to be futher polished.

Response：Thank you for your comment. The manuscript has been edited by a professional language editing service and the certificate was provided. If required, we would like further polish the manuscript.

The sentence "Previous studies suggested that protein and carbohydrate intakes, which are the main dietary components, were associated with cognition [13,14]." should be rewritten for clarity.

Response：Thank you for pointing it out. We have revised the sentence.

Page 8, lines 126-127

“As the main dietary components, protein and carbohydrate intakes were reported to associated with cognition in previous studies.”

Line 79, the word "peportedly" is inappropriately used.

Response：Thank you for pointing it out. We have corrected the word.

Reviewer #2: Overall, I have no major criticisms on this ms; however, restricted cubic splines should be mentioned in the methods and the results should be provided in the text.

Response：Thank you for your valuable comment. Restricted cubic splines were performed, and the results and figures have been added to the article.

---

## [Decision Letter · Decision Letter 2]

4 Sep 2023

Dietary fiber intake and cognitive impairment in older patients with chronic kidney disease in the United States: A cross-sectional study

PONE-D-23-15009R2

Dear Dr. liu,

We’re pleased to inform you that your manuscript has been judged scientifically suitable for publication and will be formally accepted for publication once it meets all outstanding technical requirements.

Please kindly make the changes by reviewer 2

Dietary fiber intake and cognitive impairment in older patients with chronic kidney disease in the United States: A cross-sectional study

The following points should be addressed in the proof:

1.the figures should be improved.

2.page 16, line 266: the description of “no nonlinear relationships were observed between dietary fiber intake and cognitive impairment” should be clarified further.

Kind regards,

Yee Gary Ang, MBBS MPH

Academic Editor

PLOS ONE

Reviewers' comments:

Reviewer's Responses to Questions

**Comments to the Author**

1. If the authors have adequately addressed your comments raised in a previous round of review and you feel that this manuscript is now acceptable for publication, you may indicate that here to bypass the “Comments to the Author” section, enter your conflict of interest statement in the “Confidential to Editor” section, and submit your "Accept" recommendation.

Reviewer #1: All comments have been addressed

Reviewer #3: All comments have been addressed

2. Is the manuscript technically sound, and do the data support the conclusions?

Reviewer #1: Yes

Reviewer #3: Yes

3. Has the statistical analysis been performed appropriately and rigorously? 

Reviewer #1: Yes

Reviewer #3: Yes

4. Have the authors made all data underlying the findings in their manuscript fully available?

Reviewer #1: Yes

Reviewer #3: Yes

5. Is the manuscript presented in an intelligible fashion and written in standard English?

Reviewer #1: Yes

Reviewer #3: Yes

6. Review Comments to the Author

Reviewer #1: (No Response)

Reviewer #3: Dietary fiber intake and cognitive impairment in older patients with chronic kidney disease in the United States: A cross-sectional study

The following points should be addressed in the proof:

1.the figures should be improved.

2.page 16, line 266: the description of “no nonlinear relationships were observed between dietary fiber intake and cognitive impairment” should be clarified further.

7. PLOS authors have the option to publish the peer review history of their article (what does this mean?). If published, this will include your full peer review and any attached files.

Reviewer #1: **Yes: **Hongliang Zhang

Reviewer #3: No

---

## [Editor Report · Acceptance letter]

12 Sep 2023

PONE-D-23-15009R2 

Dietary fiber intake and cognitive impairment in older patients with chronic kidney disease in the United States: A cross-sectional study 

Dear Dr. Liu:

I'm pleased to inform you that your manuscript has been deemed suitable for publication in PLOS ONE. Congratulations! Your manuscript is now with our production department. 

Kind regards, 

on behalf of

Dr. Yee Gary Ang 

Academic Editor

PLOS ONE